# Adjusting dialysis dose (Kt) scaled to Body Surface Area (BSA) could be a more logical approach

Wei Liu[1], Zhenguo Qiao[2], Yan Xu[1], Qin Zhang[1], Mengmeng Xie[1], Chunyuan Ma[1]*

1 Department of Nephrology, Suzhou Ninth People's Hospital, Suzhou Ninth Hospital Affiliated to Soochow University, Suzhou, China, 2 Department of Gastroenterology, Suzhou Ninth People's Hospital, Suzhou Ninth Hospital Affiliated to Soochow University, Suzhou, China

* patriot2020@163.com

**Data Availability Statement:** All relevant data are within the manuscript and its Supporting information files.

**Funding:** This study was financially supported by Program for the Talents in Science and Education of Wujiang District, Suzhou, China in the form of a

## Abstract

The dialysis dose, quantified as Kt, is conventionally scaled to the urea distribution volume (V) to calculate the Kt/V ratio as an indicator of dialysis adequacy. However, the body surface area (BSA) is recognized as a more accurate reflection of metabolic activity compared to V. This study presents evidence supporting the enhanced efficacy of the Kt/BSA ratio as an indicator of hemodialysis adequacy. The study population comprised 211 individuals undergoing hemodialysis, all of whom had Kt/V values determined. Body composition was assessed using bioimpedance techniques, and BSA was calculated employing the DuBois and DuBois formula. The ratio of V/BSA served as the conversion factor to derive Kt/BSA from the standard Kt/V. Participants were categorized by gender, and a comparative analysis was performed on dialysis-related parameters alongside body composition indicators. Concurrently, linear regression analysis was applied to Kt/V and Kt/BSA, as well as to pairs of V and BSA, to elucidate the relationships among these variables. The average Kt/V ratio was 1.50 with a standard deviation of 0.28. The Kt/V ratio was significantly higher in women (P < 0.01). Conversely, the Kt value, when not adjusted for body size, was significantly lower in women (P < 0.01). Upon scaling Kt to BSA to calculate Kt/BSA, the gender difference in dialysis dose adequacy was no longer statistically significant (P = 0.06). Men exhibited a significantly higher mean V/BSA ratio. Additionally, women had a slightly higher mean percentage of fat mass (P < 0.01). In contrast, women had a lower mean percentage of muscle mass (P < 0.01). Our findings suggest that the Kt/V ratio may underestimate the required hemodialysis dose for women. There was no gender difference when Kt scaling to BSA. Consequently, the Kt/BSA ratio, which accounts for body surface area, may play a significant role in more accurately scaling the hemodialysis dose.

## Introduction

Concentration-dependent toxicity is a fundamental aspect of uremia's pathogenesis [1]. The toxin concentration (C) is determined by the balance between their generation (G) and

grant (WWK202225) received by QZ. No additional external funding was received for this study.

**Competing interests:** The authors have declared that no competing interests exist.

clearance (K). In a basic single-pool model, the relationship is defined by the equation $C = \frac{G}{K}$, where C varies directly with G and inversely with K. A primary renal function is the elimination of human metabolic toxins, essential for maintaining metabolic homeostasis. The Glomerular Filtration Rate (GFR) serves as a pivotal indicator of the native kidney's clearance capacity. Given that body size and gender can affect GFR, the Estimated Glomerular Filtration Rate (eGFR) [2] has become the standard clinical measure. eGFR adjusts GFR for body surface area (BSA) using the formula $eGFR = \frac{GFR}{1.73 \times BSA}$, thereby normalizing the clearance effect across individuals of varying sizes and facilitating direct comparisons of renal function between different individuals [3]. BSA correlates closely with the production of uremic toxins that are linked to human metabolism. A larger BSA leads to increased heat loss, necessitating a higher basal metabolic rate to preserve body temperature. Although fluctuations in the toxin distribution volume (V) can affect short-term concentrations, they have minimal impact on steady-state levels. Since the generation rate (G) is assumed to scale with body size, the clearance rate (K) must be adjusted accordingly to ensure that the toxin concentration (C) remains below toxic levels.

Hemodialysis employs an artificial kidney, the dialyzer, to mimic the function of failing kidneys by removing metabolic waste products and excess water. A critical performance metric for dialyzers is urea clearance (K), which parallels GFR. The product of dialysis urea clearance (K) and dialysis session time (t), denoted as Kt, represents the volume of urea cleared in a single session. To standardize dialysis dose across patients, Kt is scaled to the urea distribution volume (V), yielding the Kt/V ratio as an indicator of dialysis adequacy [4]. Recognizing the variability in toxin removal needs, both natural and artificial kidneys consider distinct human parameters: BSA and V, respectively. However, subsequent studies [5–7] have highlighted limitations in using Kt/V as the sole measure of hemodialysis adequacy, particularly the reliance on V as the adjusting factor for Kt. The crux of the debate [8] is that V, predominantly influenced by skeletal muscle mass, may not accurately represent the generation of metabolic toxins, which also originate from the metabolic activities of internal organs. Consequently, individuals with smaller V values, such as women or less muscular men, may still necessitate adequate dialysis despite their lower V.

Normalizing GFR to BSA is routine in clinical practice. Extending this principle to dialyzer clearance by using BSA instead of V as the denominator is a logical progression. This approach to calculating dialysis dose is advantageous as it more closely aligns the function of the dialyzer with that of the native kidney, using a comparable basis for measurement. The aim of this study is to evaluate the efficacy of Kt/V in comparison with Kt/BSA, with a focus on gender differences. Our research provides evidence supporting the enhanced effectiveness of Kt/BSA as an indicator of dialysis adequacy.

## Materials and methods

### Participants and study design

This research represents a single-center, cross-sectional cohort clinical trial conducted at the Blood Purification Center of Suzhou Ninth People's Hospital in June 2023. A total of 211 patients undergoing hemodialysis and with measured Kt/V values were enrolled in the study. The inclusion criteria were set for prevalent maintenance hemodialysis patients aged over 18 years who had been on dialysis for a period exceeding three months. The exclusion criteria were defined to exclude patients with severe complications such as heart failure, malnutrition, or severe edema, those with active severe infections, individuals with neoplasms, liver cirrhosis, or who were pregnant; patients undergoing both hemodialysis and peritoneal dialysis and those with amputations. The study received ethical approval from the Suzhou Ninth People's

Hospital ethics committee (Ethical Batch No.: KYLW2023-012-01), and all participants provided informed written consent. Additionally, this study was registered as a clinical trial with the number ChiCTR2300072406, and the registration can be accessed through the website: https://www.chictr.org.cn/).

## Variables and measurement

1. Demographic and Anthropometric data

Baseline demographic and anthropometric data were extracted from the medical records of each patient. This included gender, age, height, weight, and medical history. Additionally, parameters related to dialysis were recorded, such as vascular access, dialysis vintage, blood flow rate, type of dialyzer, ultrafiltration volume, and documentation of any adverse events during dialysis, such as coagulation or premature termination.

2. Body composition parameter

Body composition was assessed using bioimpedance techniques with the Inbody S10 device from Korea Biospace. The measurements included total body water volume (TBW), body fat, muscle mass, fat-free mass (FFM), percentage of body fat, and Body Mass Index (BMI).

3. Hemodialysis treatment protocol: Hemodialysis is performed a thrice-weekly session schedule, with each session lasting four hours. Blood flow rate is set between 200–300 mL/min, and dialysate flow is 500 mL/min. A standardized, glucose-free, bicarbonate-based dialysate is used for hemodialysis procedures. Low-molecular-weight heparin calcium (LMWH-Ca) is used as an anticoagulant, The use of single-use dialyzers and dialysis tubing is mandatory to uphold the highest standards of patient safety.

4. Blood Sample Collection [4], Detection Methods, and Detection Indicators

4.1 Pro-dialysis Blood Sample Collection: 1) For patients with arteriovenous fistula (AVF), blood samples were directly collected from the arterial puncture site. 2) For patients with central venous catheter (CVC), an initial 10 mL of blood is drawn and discarded to ensure proper sampling. Subsequently, blood samples are collected for analysis, taking care to prevent dilution with heparin sealing solution.

4.2 Post-dialysis Blood Sample Collection: The ultrafiltration rate is set to zero, and the blood flow rate is gradually reduced to 100 mL/min. Blood samples are collected from the arterial end 15 to 30 seconds after this adjustment.

4.3 Blood sample Handing: To prevent the release of urea from cells into the plasma, which could affect the accuracy of the test results, serum must be promptly separated from the blood samples and sent for analysis.

4.4 Biochemistry Results: Blood samples are dispatched to our hospital's laboratory for testing. Utilizing the Siemens ADVIA 2400 automatic biochemical analyzer, various indicators are assessed, including pre-dialysis albumin and both pre- and post-dialysis urea nitrogen (BUN) levels.

5. Calculate Parameters and Formulas:

5.1 The formula for calculating the Kt/V index is as follows [4]:

$$Kt/V = -ln(R - 0.008 \times t) + (4 - 3.5 \times R) \times 0.55 \times \frac{UF}{V}$$

(where R is the ratio of post-dialysis BUN to pre-dialysis BUN, t is hemodialysis treatment time in hours, UF is ultrafiltration volume (or weight loss) for dialysis and V is body water volume).

5.2 The Kt/BSA values were calculated using the following formula [4]: where the V/BSA ratio serves as the correction factor to adjust Kt/V to Kt/BSA. The calculation of Kt/BSA

involves two steps: 1)Multiply the Kt/V value by the V/BSA ratio.2)Divide the resulting product by 20, which acts as a normalizing constant.

$$Kt/BSA = \frac{Kt/V}{20} \times \frac{V}{BSA}$$

5.3 The Body Surface Area (BSA), expressed in square meters ($m^2$), was calculated from the patient's body weight (W, in kilograms) and height (H, in centimeters), employing the DuBois and DuBois formula [9] as follows:

$$BSA = 0.007184 \times W^{0.425} \times H^{0.725}$$

(where W represents body weight in kilograms and H represents height in centimeters).

6. The National Kidney Foundation Kidney Disease Outcomes Quality Initiative (NKF-K/DOQI) guidelines [4] recommend a target single-pool Kt/V ratio of 1.4 for each hemodialysis session in patients on a thrice-weekly treatment regimen, with a minimum effective Kt/V of 1.2 delivered.

## Statistical analyses

Normally distributed data are typically presented as mean ± standard deviation ($\bar{x} \pm SD$). Student's t-test was used to compare continuous variables. Counting data were presented as the number of cases (percentage) [n (%)], with group comparisons conducted using the chi-square test ($\chi2$). Pearson's correlation was utilized to evaluate the strength and significance of the relationship between metric variables. Linear regression models the relationships between one explanatory variable and an outcome variable. The Statistical Package for the Social Sciences (SPSS), version 19 (IBM Corporation, New York, USA), was applied for all statistical analyses. A P-value < 0.05 was considered to indicate statistical significance.

## Results

### 1. The main demographic, anthropometric and dialysis characteristics

The study included 211 patients, with a mean age of 61.91 years. Diabetes mellitus was present in 33.18% of them. The average Kt/V index was (1.50±0.28), with 61.14% of participants achieving a Kt/V of at least 1.4, and 86.26% maintaining a Kt/V of at least 1.2. The main vascular access was AVF in 89.10%, and CVC in 10.90%. A detailed summary of these demographic, anthropometric, and dialysis characteristics is presented in Table 1.

### 2. Dialysis dose in relation to gender and body size

The Kt/V values were significantly higher in females (1.69 ± 0.18) than in males (1.40 ± 0.28, P < 0.01). The Kt values were significantly lower in females (47.91±5.19) than in males (53.40 ±9.76, P<0.01), and the V values were markedly reduced in females (28.62±3.33) than to males (38.80±6.65, P<0.01). Nevertheless, upon normalization of Kt to BSA, the distinction between genders was no longer significant. Females had a slightly higher average percentage of fat mass (29.38 ± 7.94) compared to males (22.92 ± 8.59, P < 0.01), yet their muscle mass percentage was lower (66.33 ± 7.20) than that of males (73.08 ± 6.93, P < 0.01). Additionally, specific anthropometric measurements, such as height, weight, body surface area (BSA), percentage of body water (V%), and muscle mass, were all considerably greater in males than in females, as detailed in Table 1.

**Table 1. Demographic, anthropometric and dialysis characteristics of 211 study patients.**

| Characteristic | All(n = 211) | female(n = 75) | male(n = 136) | $t/x^2$ | P-value |
|---|---|---|---|---|---|
| Age(y) | 61.91±13.36 | 62.85±12.38 | 61.38±13.88 | 0.77 | 0.45 |
| Diabetic(Y/N) | 70/141 | 21/75 | 49/87 | 5.35 | 0.03 |
| Vascular access(CVC/AVF) | 23/188 | 12/63 | 11/125 | 3.12 | 0.11 |
| dialysis type (HD/HDF) | 187/24 | 67/8 | 120/16 | 0.06 | 0.81 |
| t(hours) | 3.98±0.12 | 3.97±0.17 | 3.99±0.08 | -0.96 | 0.34 |
| Height(cm) | 164.06±8.99 | 155.23±4.97 | 168.93±6.71 | -16.88 | <0.01 |
| Weight(kg) | 63.88±12.64 | 55.68±8.12 | 68.40±12.43 | -8.96 | <0.01 |
| BMI(kg/m$^2$) | 23.62±3.53 | 23.11±3.18 | 23.90±3.69 | -1.58 | 0.12 |
| V (L) | 35.18±7.50 | 28.62±3.33 | 38.80±6.65 | -14.79 | <0.01 |
| V (%) | 55.3±6.06 | 51.91±5.65 | 57.18±5.45 | -6.63 | <0.01 |
| Fat mass(Kg) | 16.23±7.16 | 17.30±6.78 | 15.93±7.66 | 1.3 | 0.2 |
| Fat mass(%) | 25.22±8.90 | 29.38±7.94 | 22.92±8.59 | 5.36 | <0.01 |
| Muscle mass(Kg) | 44.96±9.59 | 36.57±4.26 | 49.59±8.51 | -14.79 | <0.01 |
| Muscle mass(%) | 70.68±7.72 | 66.33±7.20 | 73.08±6.93 | -6.67 | <0.01 |
| Kt(L) | 51.45±8.81 | 47.91±5.19 | 53.40±9.76 | -5.33 | <0.01 |
| BSA(m$^2$) | 1.69±0.19 | 1.53±0.11 | 1.78±0.17 | -12.69 | <0.01 |
| V/BSA(L/m$^2$) | 20.61±2.41 | 18.64±1.47 | 21.7±2.12 | -12.30 | <0.01 |
| Kt/V | 1.50±0.28 | 1.69±0.18 | 1.40±0.28 | 8.89 | <0.01 |
| Kt/BSA | 1.53±0.24 | 1.57±0.17 | 1.51±0.27 | 1.87 | 0.06 |

Note: central venous catheter(CVC), arteriovenous fistula(AVF), hemodialysis(HD), hemodiafiltration(HDF), hemodialysis treatment time(t), Body Mass Index(BMI), Volume of urea cleared(Kt), Urea distribution volume(V), Body surface area (BSA)

## 3. Relationship between Kt/V and Kt/BSA by gender

Fig 1: A scatter plot illustrates the distribution of Kt/V and Kt/BSA values, differentiated by gender, with superimposed lines representing the optimal linear regression fits for each gender. Female, Kt/V and Kt/BSA regression equation: $Kt/BSA_{(f)} = 0.40 + 0.69 \times Kt/V_{(f)}$ (Analysis of variance on the model result: F = 101.32, P<0.01; r = 0.76, $R^2$ = 0.58, Std. Error of the Estimate is 0.11, Durbin-Watson Statistic is 2.19); Male, Kt/V and Kt/BSA regression equation: $Kt/BSA_{(m)} = 0.31 + 0.86 \times Kt/V_{(m)}$ (Analysis of variance on the model result: F = 406.41, P<0.01; r = 0.87, $R^2$ = 0.75, Std. Error of the Estimate is 0.14, Durbin-Watson Statistic is 1.72). The linear regression analysis of the scatter plot yields nearly parallel lines, with the male regression line positioned above the female line. It is important to observe that at equivalent Kt/V levels, females consistently present a higher Kt/BSA ratio.

## 4. Relationship between V and BSA by gender

Fig 2: This figure presents a scatter plot that contrasts water volume (V) with Body Surface Area (BSA), with the data stratified by gender. Female, BSA and V regression equation: $V_{(f)} = -5.01 + 21.92 \times BSA_{(f)}$ (Analysis of variance on the model result: F = 90.58, P<0.01; r = 0.74, $R^2$ = 0.55, Std. Error of the Estimate is 2.24, Durbin-Watson Statistic is 2.19); Male, BSA and V regression equation: $V_{(m)} = -22.29 + 34.91 \times BSA_{(m)}$ (Analysis of variance on the model result: F = 418.19, P<0.01; r = 0.87, $R^2$ = 0.76, Std. Error of the Estimate is 3.29, Durbin-Watson Statistic is 1.95). The regression lines for males and females intersect, indicating a difference in the relationship between V and BSA between gender. The slope of the regression line for males is significantly steeper than that for females, suggesting a more pronounced increase in water

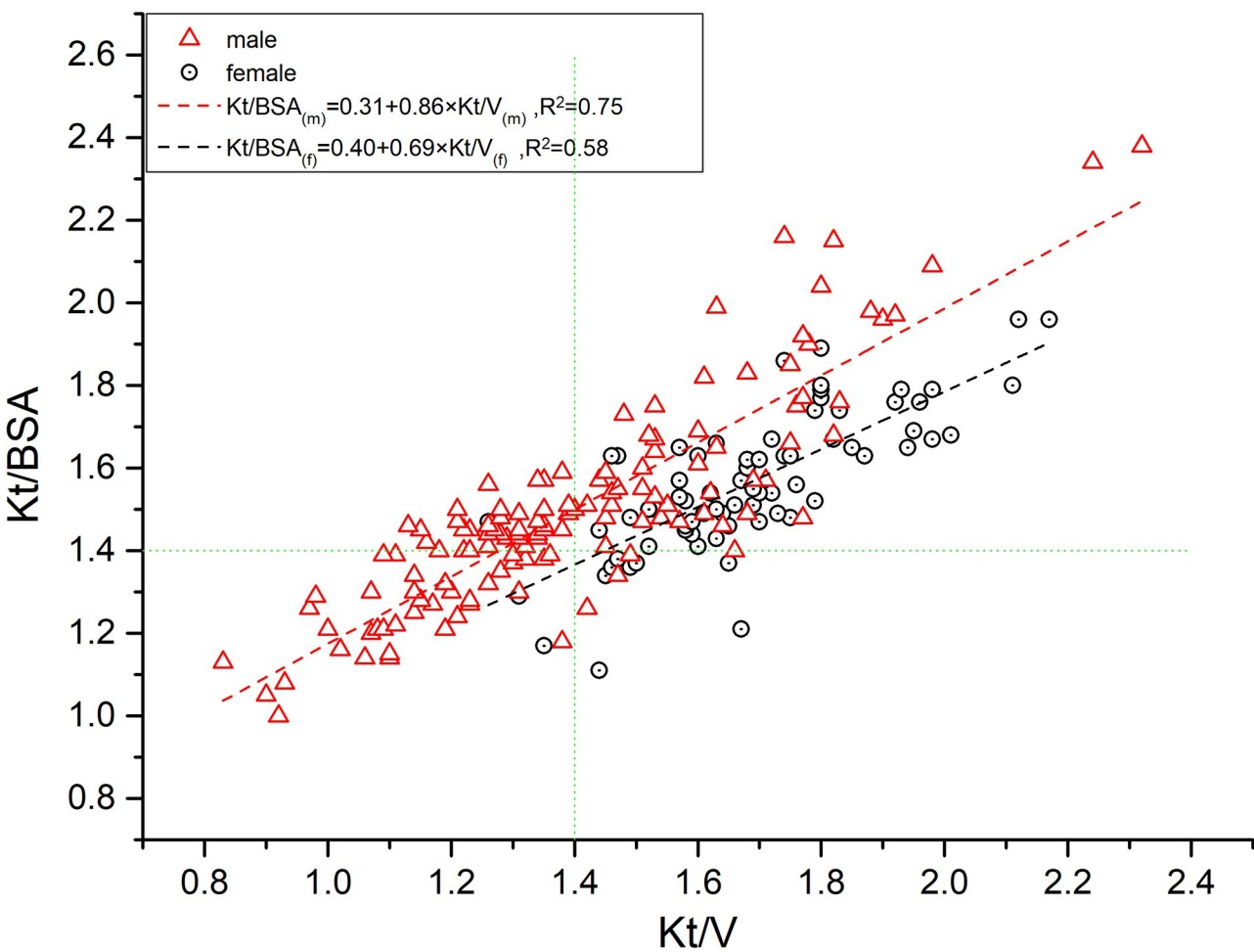

**Fig 1. Relationship between Kt/V and Kt/BSA by gender.**

volume relative to BSA in males. Consequently, at equivalent BSA values, females generally have a lower water volume (V) compared to males.

## Discussion

Adequate hemodialysis is essential for enhancing patients' quality of life, minimizing complications, and improving prognosis. Precise evaluation of dialysis adequacy is critical. Broadly defined, it involves the effective removal of water and uremic toxins through dialysis treatment, along with efficient management of various complications. Patients should experience comfort during dialysis, which can improve their quality of life and enable social participation [10]. More specifically, dialysis adequacy focuses on the removal of small molecular solutes, such as urea [4].

Urea, a primary end-product product of protein metabolism, is a key small molecular weight solute uniformly distributed throughout the body. Its rapid turnover rate makes it efficiently removable by dialysis. The urea clearance index (Kt/V), derived from urea kinetic models [11, 12], is widely recognized for evaluating dialysis adequacy due to its reliability and cost-effectiveness in clinical settings. Kt/V reflects the ratio of the volume of fluid purified to the

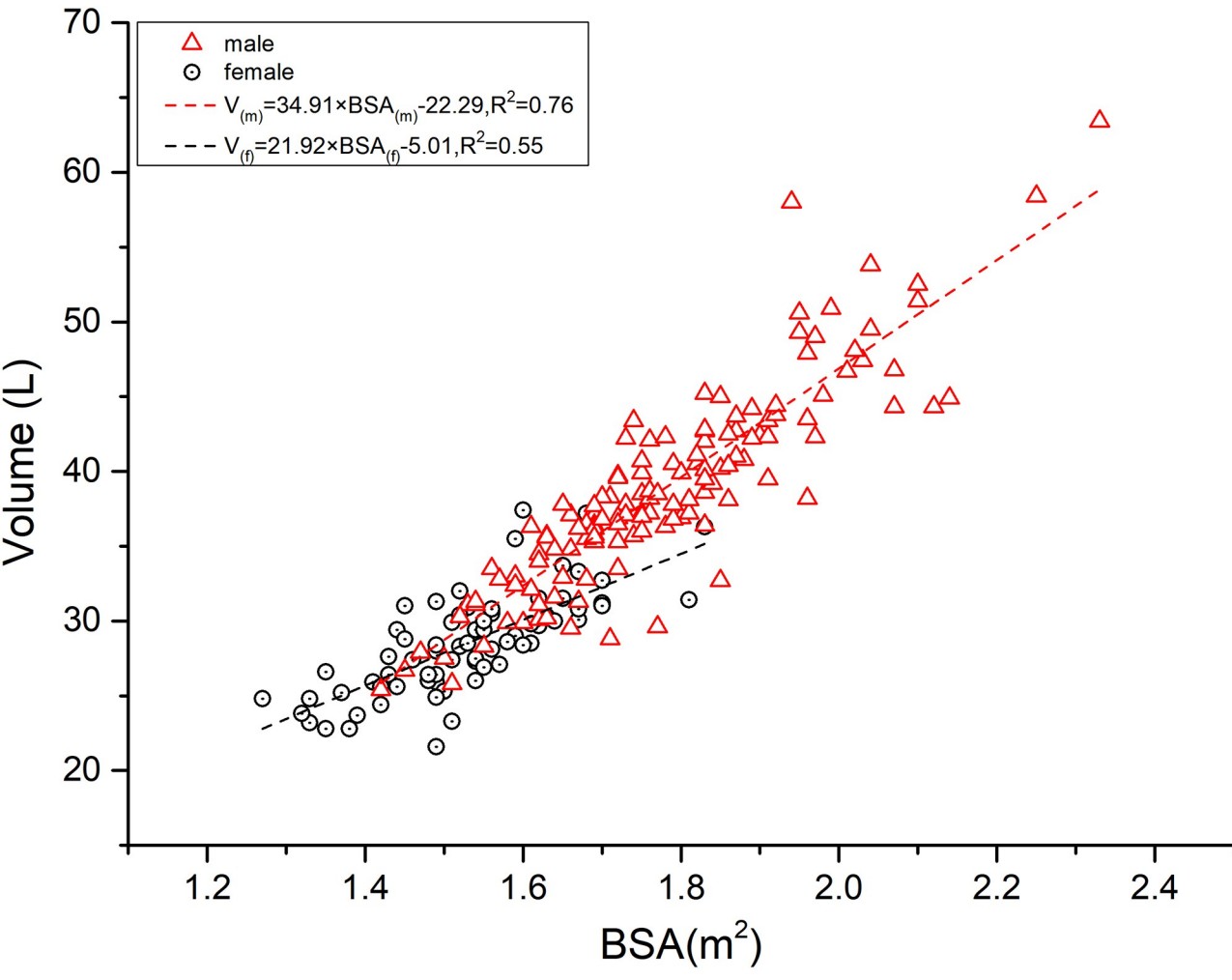

**Fig 2. Relationship between V and BSA by gender.**

total volume of urea distribution during a dialysis session, where K is the urea clearance rate, t is the dialysis time, and V approximates total body water content.

Observational studies [13–16] have shown a correlation between Kt/V and mortality, suggesting that higher dialysis doses may improve patient survival. However, other studies [17, 18] did not find a survival benefit at higher Kt/V levels. Furthermore, a "U-shaped" or "reverse J-shaped" relationship between Kt/V and mortality has been identified [19, 20], indicating that excessively high Kt/V levels, beyond a certain threshold (e.g. spKt/V > 1.3), may not further decrease the risk of death and could potentially increase it.

The single-pool urea kinetic model, while widely used, has limitations. It assumes uniform urea distribution, constant production, and clearance following first-order kinetics. However, Kt/V is derived from urea levels before and after dialysis and can be influenced by various factors [20, 21], including sampling methods and dialysis parameters. Additionally, Kt/V may not accurately reflect the clearance of larger molecular weight toxins with different kinetics.

The urea distribution volume (V) in the model often does not consider the unique characteristics of uremic patients, such as the distinct body fluid distribution characteristics and nutritional status. Viewing V simply as the distribution volume of passive, pure, and

homogeneous urea overlooks the structural information about the human body that V inherently carries. A low V, common in the elderly and malnourished individuals, is associated with a higher risk of mortality [22–25]. Given that the V acts as an independent predictor of patient mortality, adjusting dialysis dose based on V could complicate the observed relationship between dialysis dose and mortality. This adjustment may introduce complex and not always predictable consequence.

Increasing evidence [8, 23] indicates that certain patient cohorts, including females, petite males, those suffering from malnutrition, and the elderly, potentially face the risk of suboptimal dialysis when evaluated against conventional Kt/V target values. This predominantly stems from the reduced muscle mass proportion within these demographics, culminating in a comparatively diminished V value. The determination of the Kt/V metric is contingent upon both the urea clearance (Kt) and the urea distribution volume (V). A low V value can yield an elevated Kt/V metric, a circumstance that does not invariably signify either excessive dialysis or its sufficiency. Consequently, Kt/V might under-predict dialysis adequacy for females and diminutive males [8], thereby leading to lesser dialysis prescriptions. In contrast, corpulent patients may manifest an augmented V value, as computed relative to body mass, potentially causing Kt/V to under-predict dialysis sufficiency owing to the metabolic lethargy and reduced aqueous content characteristic of adipose tissue. This phenomenon also constitutes a salient rationale elucidating the "obesity-survival paradox" [26, 27] observed among dialysis recipients.

In response to the limitations of the Kt/V metric, researchers have initiated the exploration of additional metrics to evaluate dialysis adequacy. The Body Surface Area (BSA) is commonly utilized in clinical settings as a scaling factor for various physiological parameters, including basal metabolic rate [28], the left ventricular mass index [29], medication doses [30], cardiac output [31], and the Glomerular Filtration Rate (GFR) [3]. A comparative study of the native renal GFR, measured by iothalamate clearance in potential kidney donors, demonstrated that normalization of GFR for BSA eliminates gender disparities; in contrast, normalization by the anthropometric volume (V) does not achieve this uniformity [3]. Women tend to have a lower V/BSA ratio compared to men, which is attributed to their higher proportion of adipose tissue —having a lower water content—versus the water-rich muscle tissue. The practice of standardizing dialysis dose (Kt) by BSA is a logical advancement, considering that the production rate of uremic toxins is a critical factor in determining dialysis requirements for patients with different body compositions. Under the steady state provided by regular dialysis, toxin concentrations are assumed to be directly proportional to their generation rates and inversely proportional to their clearance rates. Research by Lowrie [32] suggests that BSA provides a more accurate reflection of metabolic activity than V, thus better representing the generation of uremic toxins.

Under conditions of identical height and weight, the BSA tends to have similar values among males and females. However, the total body water (V) is generally higher in males due to their greater muscle mass. Currently, there is no strong evidence to suggest that muscle tissue significantly contributes to the production of uremic toxins. Therefore, if dialysis dose is scaled to V, males may be prescribed a more intensive dialysis regimen compared to females with a similar BSA. The pathophysiology of uremia is more closely related to the generation rate of uremic toxins than to the volumetric distribution of urea. This generation rate is likely more closely associated with metabolic processes. Metabolic activity typically increases in individuals with a larger BSA. Scaling Kt to BSA aligns with normal physiology, as the production of metabolic waste should correspond to total energy expenditure. Since BSA is more significantly influenced by stature than by mass, using BSA rather than V in the denominator can reduce inaccuracies when a patient experiences weight loss or retains edematous fluid. Such

changes may not directly affect the need for dialysis but could cause significant variations in the Kt/V ratio. In cases where individuals of the same weight but different heights are compared, the taller individual invariably has a faster metabolism due to their larger BSA. This is because taller individuals have a greater BSA and thus require a faster metabolic rate to counteract heat dissipation. It is logical that individuals with a higher basal metabolic rate would require a more extensive dialysis prescription. BSA shows a stronger correlation with both the basal metabolic rate and body configuration across genders than the urea distribution volume (V). Moreover, the Kt/BSA metric has shown a stronger association with survival outcomes compared to Kt/V [33]. Tailoring the dialysis dose (Kt) according to BSA may provide additional benefits for females and smaller patients.

## Conclusions

Our study has illuminated that the Kt/V ratio, a standard measure of dialysis efficacy, is influenced by both dialysis dose (Kt) and urea distribution volume (V). We found that a lower V can unexpectedly result in a higher Kt/V ratio, which does not necessarily equate to either excessive or adequate dialysis. The body compositions of men and women differ significantly, with women typically having a lower muscle mass and a higher fat mass relative to body weight. This results in women having a lower V and V% compared to men. Contrary to initial appearances, the Kt/V ratio, while higher in women, does not indicate superior dialysis adequacy when the dialysis dose (Kt) is actually lower in women than in men. By reassessing dialysis adequacy with the Kt/BSA ratio, which accounts for body surface area, the perceived advantage in dialysis adequacy for women is diminished. BSA is more closely aligned with metabolic activity and urea production, and its correlation with body weight is more robust than that of V. Thus, the Kt/BSA ratio may play a pivotal role in adjusting dialysis dose and merits further exploration for its clinical utility.

The study, however, is not without limitations. The clearance of small molecules (K) was ascertained using the online ionic clearance (OLC) method and calculated over the treatment time (t) to determine the total dose (Kt). This method-derived Kt/V value is unaffected by various factors such as vascular access recirculation, cardiopulmonary recirculation, urea nitrogen rebound, sampling errors, laboratory inconsistencies, residual renal function, etc., providing an accurate reflection of the hemodialysis impact on urea clearance. Nonetheless, future prospective clinical trials are essential to evaluate the long-term effects of different Kt/BSA levels on survival, hospitalization, and complication rates, thereby substantiating the rationale and efficacy of using Kt/BSA as a measure of dialysis adequacy.

## Supporting information

**S1 Data.**
(XLSX)

## Author Contributions

**Conceptualization:** Chunyuan Ma.

**Data curation:** Yan Xu.

**Formal analysis:** Qin Zhang.

**Investigation:** Mengmeng Xie.

**Supervision:** Zhenguo Qiao.

**Writing – original draft:** Wei Liu.

**Writing – review & editing:** Chunyuan Ma.

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
