## [Decision Letter · Decision Letter 0]

22 Aug 2024

PONE-D-24-24065Adjusting dialysis dose (Kt) scaled to Body Surface Area (BSA) could be a more logical approachPLOS ONE 

Dear Dr. ma,

thank you for submitting your manuscript to PLOS ONE. After careful consideration, we feel that it has merit but does not fully meet PLOS ONE’s publication criteria as it currently stands. Therefore, we invite you to submit a revised version of the manuscript that addresses the points raised during the review process. 

We look forward to receiving your revised manuscript.

Kind regards,

Diego Moriconi

Academic Editor

PLOS ONE

2. In the online submission form, you indicated that [The datasets used and/or analyzed during the present study are available from the corresponding author on reasonable request.]. 

Additional Editor Comments:

Dear Dr. Chunyuan Ma,

we have received the reports from our advisors on your manuscript, "Adjusting dialysis dose (Kt) scaled to Body Surface Area (BSA) could be a more logical approach", submitted to our journal Plos One.

Based on the advice received, I have decided that your manuscript could be reconsidered for publication should you be prepared to incorporate minor revisions.

When preparing your revised manuscript, you are asked to carefully consider the reviewer comments which can be found below, and submit a list of responses to the comments.

Best regards,

Dr. Diego Moriconi, MD, PhD.

Reviewers' comments:

Reviewer's Responses to Questions

**Comments to the Author**

1. Is the manuscript technically sound, and do the data support the conclusions?

Reviewer #1: Yes

Reviewer #2: Yes

2. Has the statistical analysis been performed appropriately and rigorously? 

Reviewer #1: Yes

Reviewer #2: N/A

3. Have the authors made all data underlying the findings in their manuscript fully available?

Reviewer #1: Yes

Reviewer #2: Yes

4. Is the manuscript presented in an intelligible fashion and written in standard English?

Reviewer #1: No

Reviewer #2: Yes

5. Review Comments to the Author

Reviewer #1: The manuscript presents an interesting approach to scaling dialysis dose (Kt) to body surface area (BSA) instead of the conventional urea distribution volume (V) for a more accurate reflection of metabolic activity. The study is well-structured, with a clear abstract and comprehensive results.

The manuscript states that linear regression analysis was applied, but it does not provide details about the regression model assumptions, the goodness-of-fit metrics, or the potential multicollinearity issues. Including information about the model diagnostics (residual analysis, R-squared values) and any steps taken to validate the regression model would strengthen the analysis.

Reviewer #2: In this single-centre cross-sectional study, the researchers investigated the efficacy of hemodialysis, as measured by Kt/V versus Kt/BSA, in 211 patients with end-stage chronic kidney disease undergoing chronic hemodialysis. The focus of the two dialysis parameters for adequacy was on gender comparison. The inclusion and exclusion criteria are clear. The study was approved by the ethics committee and all participants signed a written informed consent. Most of the participants had an arteriovenous fistula as vascular access. The researchers found significantly higher Kt/V values in women than in men and after normalisation from Kt to BSA, the difference between the two sexes was no longer statistically significant.

The manuscript is well written, the entire text needs to be reviewed by a native English speaker.

I have the following comments, questions, suggestions:

1. When were the blood samples for the Kt/V measurements taken? At the first, second or third hemodialysis session of the week?

2. Do all patients have the same weekly schedule for dialysis sessions?

3. Data on the duration of the hemodialysis session of the participants at the time of blood sampling is missing.

4. Data on the type of dialysis (hemodialysis, hemofiltration, hemodiafiltration) are missing.

5. Minor comments:

Table 1: SBA is incorrect, it is BSA

Relationship between Kt/V and Kt/BSA by gender (not by sex)

6. PLOS authors have the option to publish the peer review history of their article (what does this mean?). If published, this will include your full peer review and any attached files.

Reviewer #1: No

Reviewer #2: No

---

## [Author Response · Author response to Decision Letter 0]

30 Aug 2024

Response to reviewers

Dear editor and reviewers of PLoS One:

Our reference: PONE-D-24-24065

Title: Adjusting dialysis dose (Kt) scaled to Body Surface Area (BSA) could be a more logical approach

By: Liu Wei et al

Thank you very much for your letter and for the editors’ and reviewers’ comments concerning our manuscript entitled “Adjusting dialysis dose (Kt) scaled to Body Surface Area (BSA) could be a more logical approach” (ID: PONE-D-24-24065). These comments are of great reference value to the revision and improvement of our paper and have important guiding significance to our researches. The main corrections in the paper and the responds to the reviewer’s comments are as flowing:

Reviewer #1: The manuscript presents an interesting approach to scaling dialysis dose (Kt) to body surface area (BSA) instead of the conventional urea distribution volume (V) for a more accurate reflection of metabolic activity. The study is well-structured, with a clear abstract and comprehensive results.

The manuscript states that linear regression analysis was applied, but it does not provide details about the regression model assumptions, the goodness-of-fit metrics, or the potential multicollinearity issues. Including information about the model diagnostics (residual analysis, R-squared values) and any steps taken to validate the regression model would strengthen the analysis.

Author's reply：The article adds the corresponding content as follows:

“Femle，Kt/V and Kt/BSA regression equation：Kt/BSA(f)=0.34+0.73×Kt/V(f)(Analysis of variance on the model result:F=107.59,P<0.01; r=0.77,R2=0.60,Std. Error of the Estimate is 0.11,Durbin-Watson Statistic is 2.15);Male，Kt/V and Kt/BSA regression equation：Kt/BSA(m)=0.30+0.87×Kt/V(m)(Analysis of variance on the model result:F=384.70,P<0.01; r=0.86,R2=0.74,Std. Error of the Estimate is 0.14,Durbin-Watson Statistic is 1.93)”

and

”Femle，BSA and V regression equation：V(f)=-5.01+21.92×BSA(f)(Analysis of variance on the model result:F=90.58 ,P<0.01 ; r=0.74,R2=0.55,Std. Error of the Estimate is 2.24 ,Durbin-Watson Statistic is 2.19) ;Male，BSA and V regression equation：V(m)=-22.29+34.91×BSA(m) (Analysis of variance on the model result:F=418.19 ,P<0.01 ; r=0.87,R2=0.76,Std. Error of the Estimate is 3.29,Durbin-Watson Statistic is 1.95).”

Reviewer #2: In this single-centre cross-sectional study, the researchers investigated the efficacy of hemodialysis, as measured by Kt/V versus Kt/BSA, in 211 patients with end-stage chronic kidney disease undergoing chronic hemodialysis. The focus of the two dialysis parameters for adequacy was on gender comparison. The inclusion and exclusion criteria are clear. The study was approved by the ethics committee and all participants signed a written informed consent. Most of the participants had an arteriovenous fistula as vascular access. The researchers found significantly higher Kt/V values in women than in men and after normalisation from Kt to BSA, the difference between the two sexes was no longer statistically significant.

The manuscript is well written, the entire text needs to be reviewed by a native English speaker.

I have the following comments, questions, suggestions:

1.When were the blood samples for the Kt/V measurements taken? At the first, second or third hemodialysis session of the week?

Author's reply：In the article, Kt/V represents the dose for a single treatment,The blood samples were taken randomly.The measurement time is not fixed.

2.Do all patients have the same weekly schedule for dialysis sessions?

Author's reply：Yes,the hemodialysis regimen mandates a thrice-weekly session schedule, with each session lasting four hours.

The article added "Hemodialysis Treatment Protocol" related content as follows:

3.“3.Hemodialysis Treatment Protocol: Hemodialysis is performed a thrice-weekly session schedule, with each session lasting four hours. Blood flow rate is set between 200-300 mL/min, and dialysate flow is 500 mL/min. A standardized, glucose-free, bicarbonate-based dialysate is used for hemodialysis procedures. Anticoagulation with calcium-based heparin prevents clots, The use of single-use dialyzers and dialysis tubing is mandatory to uphold the highest standards of patient safety.”

3.Data on the duration of the hemodialysis session of the participants at the time of blood sampling is missing.

Author's reply：The article adds the corresponding content as follows: please see table 1

4.Data on the type of dialysis (hemodialysis, hemofiltration, hemodiafiltration) are missing.

Author's reply：We only have two types of dialysis(hemodialysis(HD) and hemodiafiltration(HDF)).The article adds the corresponding content as follows: please see table 2

5. Minor comments:

Table 1: SBA is incorrect, it is BSA

Relationship between Kt/V and Kt/BSA by gender (not by sex)

Author's reply：The corresponding part of the article has been revised.

Once again, thank you very much for your comments and suggestions.

2024-08-30

---

## [Decision Letter · Decision Letter 1]

16 Sep 2024

Adjusting dialysis dose (Kt) scaled to Body Surface Area (BSA) could be a more logical approach

PONE-D-24-24065R1

Dear Dr. Chunyuan Ma,

We’re pleased to inform you that your manuscript has been judged scientifically suitable for publication and will be formally accepted for publication once it meets all outstanding technical requirements.

Kind regards,

Diego Moriconi

Academic Editor

PLOS ONE

Additional Editor Comments (optional):

Dear Dr. Chunyuan Ma,

we have received the report from our advisor on your manuscript, "Adjusting dialysis dose (Kt) scaled to Body Surface Area (BSA) could be a more logical approach", submitted to our journal Plos One.

Based on the advice received, the paper is considered, in light of the improvements made, suitable for publication.

Best regards,

Dr. Diego Moriconi, MD, PhD.

Reviewers' comments:

Reviewer's Responses to Questions

**Comments to the Author**

1. If the authors have adequately addressed your comments raised in a previous round of review and you feel that this manuscript is now acceptable for publication, you may indicate that here to bypass the “Comments to the Author” section, enter your conflict of interest statement in the “Confidential to Editor” section, and submit your "Accept" recommendation.

Reviewer #2: All comments have been addressed

2. Is the manuscript technically sound, and do the data support the conclusions?

Reviewer #2: Yes

3. Has the statistical analysis been performed appropriately and rigorously? 

Reviewer #2: Yes

4. Have the authors made all data underlying the findings in their manuscript fully available?

Reviewer #2: Yes

5. Is the manuscript presented in an intelligible fashion and written in standard English?

Reviewer #2: Yes

6. Review Comments to the Author

Reviewer #2: The authors took into account all my suggestions and comments and I think that the paper has been improved. Thank you.

I think the article is now suitable for publication.

7. PLOS authors have the option to publish the peer review history of their article (what does this mean?). If published, this will include your full peer review and any attached files.

Reviewer #2: No

---

## [Editor Report · Acceptance letter]

27 Sep 2024

PONE-D-24-24065R1 

PLOS ONE

Dear Dr. Ma, 

I'm pleased to inform you that your manuscript has been deemed suitable for publication in PLOS ONE. Congratulations! Your manuscript is now being handed over to our production team.

Kind regards, 

on behalf of

Dr. Diego Moriconi 

Academic Editor

PLOS ONE